# Eco-Friendly Multilayer Coating Harnessing the Functional Features of Curcuma-Based Pigment and Rice Bran Wax as a Hydrophobic Filler

**DOI:** 10.3390/ma16227086

**Published:** 2023-11-08

**Authors:** Massimo Calovi, Stefano Rossi

**Affiliations:** Department of Industrial Engineering, University of Trento, Via Sommarive 9, 38123 Trento, Italy; massimo.calovi@unitn.it

**Keywords:** curcuma, rice bran wax, environmentally friendly additive, wood paint, green coating

## Abstract

This work aims to highlight the multiple features shown by curcuma-based pigment and rice bran wax, which can be selectively employed as bio-based additives for the realization of multilayer wood coatings with multiple functionalities, harnessing the capabilities of the two environmentally friendly fillers, in line with current environmental sustainability trends. The role of the two green materials on the morphology of the composite layers was examined through observations employing scanning electron and optical microscopy, revealing a strong alteration of the film’s appearance, both its color and reflectivity. Additionally, their influence on the paint’s resilience was assessed by exposing the samples to UV-B radiation and consecutive thermal shocks. The coating displayed a clear and uniform change in color because of substantial curcuma powder photo-degradation but it remained exceptionally stable when subjected to thermal stresses. Moreover, the protective properties of the coatings were evaluated by conducting liquid resistance tests and water uptake tests, while the hardness and the abrasion resistance of the coatings were assessed to evaluate the effect of the additives on the mechanical properties of the coatings. In conclusion, this study showcases the promising joint action of curcuma-based pigment and rice bran wax in multilayer coatings. This combination offers vibrant yellow tones and an appealing appearance to the paint, enhances the surface’s water-repellent properties, and improves the mechanical resistance of the coatings.

## 1. Introduction

Wood has been a widely exploited resource by humans throughout history [1], owing to its unique physical and chemical characteristics [2]. These properties include an impressive ratio of strength to weight [3] and simplicity in handling [4]. Additionally, today, wood is highly valued due to its widespread availability and its simplicity as a material, along with its distinct aesthetic qualities [5]. Nonetheless, because of its lignocellulosic composition, wood is susceptible to issues such as flammability [6], deterioration resulting from humidity [7], and harm from sunlight exposure [8]. To tackle these real-world issues, wooden parts are frequently coated with organic layers, enhancing their durability by providing protection against solar radiation [9], changes in moisture levels [10], chemical assaults [11], physical damage [12], and the prevention of the proliferation of detrimental microorganisms such as fungi [13,14].

The extensive use of wood in outdoor settings has spurred interest from both the academic and industrial fields to investigate novel methods for enhancing the functionalities of wood coatings [15]. One such approach involves the modification of wood coatings’ UV absorption capabilities through the incorporation of various nanoparticles, including TiO_2_ [16], ZnO [17], SiO_2_ [18], and CeO_2_ [19]. Likewise, advanced nanostructures known for their outstanding hardness, stiffness, and heat resistance, such as nanosilica [20], nanoalumina [21], nanoclay [22], and nanocellulose [23], have been employed to improve the mechanical characteristics and moisture resistance of wood protective layers [24]. Furthermore, the ability to combat bacteria and fungi of wood coatings has been bolstered by incorporating nanomaterials like copper nanopowders [25], nanotitanium [26], and silver [27].

However, the use of colored paint to impart specific aesthetic effects to wooden products, utilizing new colorants [28] and unique shine levels [29], represents a growing fashion in the wood protective coatings market [30]. In this regard, the durability of these colorants in wood coatings has garnered significant attention in recent research [31]. It is essential to keep in mind that when introducing novel colorants, they should provide distinct visual effects while maintaining the protective characteristics of the organic coating. The combination of wood coatings featuring various pigment varieties can potentially lead to significant issues, such as a reduction in the protective efficacy of the organic coating due to the introduction of discontinuities in the polymeric matrix or concerning the low inherent durability of the same pigments [32].

The wood coating sector is presently adopting two notable and crucial focal points: eco-friendly materials and the concept of a circular economy. As the industry progressively explores environmentally friendly and versatile substitutes for traditional synthetic additives [33], which often lack considerations for environmental sustainability during production [34], scientific investigation is delving into the utilization of organic supplements in coatings [35]. In pursuit of this goal, scientists have recently studied the impacts of integrating a range of components into wood coatings, including linseed oil [36], cellulose fibers [37], colorants derived from wood byproducts [38], microbial coloring [39], and dyes extracted from fungi [40] and microalgae [41,42].

From this standpoint, curcuma powder and rice bran wax-based fillers have the potential to serve as intriguing multifunctional additives for wood coatings.

*Curcuma L.*, a genus belonging to the *Zingiberaceae* family, commonly known as turmeric, comprises a collection of perennial rhizomatous herbs that naturally occur in tropical and subtropical regions. The cultivation of Curcuma plants is widespread in Asia, Australia, and South America, primarily in tropical and subtropical areas [43]. The number of accepted Curcuma species is approximately 93–100, although the exact count remains a topic of debate [44]. The genus is most celebrated for its vital role in providing color and flavor enhancements in Asian culinary traditions, as well as its contributions to traditional medicine, spices, dyes, fragrances, cosmetics, and decorative plants [45]. The rhizome, which is the commonly used part of the plant, holds the primary active components. These include the nonvolatile curcuminoids and the volatile oil [46]. Specifically, curcuminoids, which encompass curcumin, demethoxycurcumin, and bisdemethoxycurcumin, are polyphenolic derivatives of curcumin discovered in the rhizome. They are nontoxic compounds and exhibit a broad spectrum of biological activities [47]. These are the main coloring compounds [48,49] of curcuma, capable of giving an intense yellow color to various substances. Thus, curcumin can effectively serve as the vibrant yellow tri-color element for applying non-harmful, bio-based organic dyes to wood coatings [50].

Conversely, rice bran wax is a natural wax obtained from the husk or bran of rice (*Oryza sativa*), which is a by-product of rice production. As a staple food for over half of the global population, rice is among the most extensively cultivated and processed food crops worldwide [51]. Rice bran wax is composed of renewable hydrocarbon compounds and it is obtained through a dewaxing process during the refinement of rice bran oil [52,53], which involves stages like degumming, deacidification, bleaching, deodorization, and winterization [53,54]. Due to its inherent hydrophobic properties, rice bran wax has the potential to serve as a plentiful and cost-effective bio-based filler, enhancing the protective capabilities of organic coatings. The low surface free energy of wax [55,56] can indeed be harnessed to enhance the hydrophobic properties of wood, as suggested by recent studies [57].

Hence, taking into account the crucial factors of incorporating in wood coatings both pigments and reinforcing fillers which align with environmental sustainability and the principles of the circular economy, this study seeks to develop a multilayer wood coating that harnesses the color properties of curcuma and the hydrophobic characteristics of rice bran wax. The study examines the visual and versatile characteristics offered by the two environmentally friendly materials and also investigates their impact on the durability of the polymeric coating.

To assess the influence of the two green products (curcuma powder and rice bran wax) on the various characteristics of a wood coating, they were added to a commercially available waterborne wood paint. The impacts of the bio-based additives were examined through various characterization methods, including optical stereomicroscope analysis, colorimetric measurements, and gloss analysis. These evaluations were conducted to examine how the fillers influenced the coatings’ morphology, general structure, and visual attributes. Additionally, the coatings’ endurance was assessed through accelerated degradation trials, which included exposure to environmental conditions in a climatic chamber and exposure to UV-B radiation. FTIR measurements were carried out to detect any chemical alterations in the coatings due to UV-B radiation exposure, while alterations in visual characteristics were tracked through colorimetric and gloss examinations. Likewise, colorimetric measurements were utilized to investigate any potential decay of the coating provoked by the two green additives during exposure in the climatic chamber. The performance of the coatings was further studied through visual inspections and optical microscope observations. The adhesion of the coating was assessed through the cross-cut test. Additionally, to evaluate the potential role of the two natural additives on the paint’s protective qualities, the performance of the coatings was analyzed via tests for chemical resistance, wettability, and liquid water absorption. These assessments also involved an examination of potential alterations in color. The mechanical attributes of the coatings, as well as the effect of the green fillers, were appraised through the Buchholz hardness indentation test and the scrub test. These tests offered insights into the coatings’ toughness and resilience and helped elucidate the impact of the bio-based additives on their performance.

## 2. Materials and Methods

### 2.1. Materials

The curcuma powder and the rice bran wax were provided by Garzanti Specialties (Milan, Italy) and Azelis Italia (Milan, Italy), respectively, and employed as received. The matured rhizomes of *Curcuma longa L*. were used to obtain the powders, which were then dried, cleaned, ground, screened, and checked for any metal contaminants. The manufacturing company subjected the product to a steam sanitization process, carefully controlling the pressure, temperature, and duration of treatment. The final curcuma powder possesses granules of dimension 100% through 40 mesh (<420 µm) with a volumetric mass of 0.60 g/mL and it is primarily composed of ash (≤10.0%), curcumin (≥3.0%), and insoluble acid ash (≤2.5%), as specified by the supplier. The rice bran wax *NatureFine R331* was supplied as micro-particles with an average size of 6.0–10.0 µm and a density of 0.97 g/mL. The material possesses a melting point of 77–82 °C and an acid value < 12. The poplar wood substrates, measuring 150 × 150 × 2 mm^3^, were sourced from Cimadom (Lavis, Italy). The waterborne acrylic bio-based paint TECH20 was provided by ICA Group (Civitanova Marche, Italy). The formulation of this product incorporates raw materials that are derived from renewable sources. The paint has a specific weight of 1.01–1.18 g/mL and a viscosity of 50–60 s Ford Cup 5. Sodium chloride (≥99.0%) and ethanol (99.8%) were purchased from Sigma-Aldrich (St. Louis, MO, USA) and used as received. The commercial detergent disinfectant product Suma Bac D10 Cleaner and Sanitiser (Diversey—Fort Mill, SC, USA), containing benzalkonium chloride (3.0–10.0 wt.%), and the cataphoretic red ink Catafor 502XC (Arsonsisi, Milan, Italy) were purchased and used for the liquid resistance tests.

### 2.2. Samples Production

Prior to applying the paint, the poplar wood panels were subjected to a pre-painting procedure using 320-grit sandpaper to achieve an even and polished surface. Afterward, the creation of the four sets of samples described in Table 1 was carried out by applying multiple coats of paint using a spraying method. The paint was applied using the guidelines provided by the supplier, employing a pressure level of 3 bar and an application rate of 100 g per square meter. In some cases, the paint was supplemented with the two bio-based fillers to impart specific characteristics to each layer. After applying each individual layer, a period of 4 h was allowed for air drying at room temperature before applying the next coating.

Sample TT served as a reference point and was created by applying two coats of the commercial bio-based transparent paint (T). Likewise, sample CC comprises two layers; but, in this instance, the paint was mixed with 5 wt.% of curcuma powder (C) to achieve a vibrant yellow hue. The concentration of curcuma, set at 5 wt.%, was chosen to achieve a bright yellow color that is sufficiently intense but still allows the distinctive veined texture of the wooden substrate to remain visible without being completely covered. Due to the susceptibility of the curcuma-based pigment to photo-degradation [58,59,60] and potential deterioration when exposed to liquids [61,62], an extra layer of pure transparent paint (T) was applied to sample CC, creating an extended sample defined as CCT. This additional layer served as a protective measure for the durability of the curcumin pigment. Finally, to introduce a new functionality to sample CCT, sample CCW was developed. In this case, the third layer was enriched with 10 wt.% of rice bran wax (W) to enhance the system’s hydrophobic and protective properties. The amount of rice bran wax was selected based on a previous work that exploited carnauba wax to improve the barrier properties of wood paints [41]. Each paint mixture employed in the samples’ deposition was mechanically blended for 30 min prior to application.

### 2.3. Characterization

The low vacuum scanning electron microscope (SEM) JEOL IT 300 (JEOL, Akishima, Tokyo, Japan) was employed to investigate the characteristics of the two bio-based additives and examine the exterior and inner structure of the layers. The objective was to assess if the curcuma and wax powders affected the structural morphology of the layers. Energy-dispersive X-ray spectroscopy (EDXS, Bruker, Billerica, MA, USA) was conducted to determine the elemental constituents of the curcuma and rice bran wax. The appearance of the coatings was assessed using the optical stereomicroscope Nikon SMZ25 (Nikon Instruments Europe, Amstelveen, The Netherlands) and by means of colorimetric analysis carried out with a Konica Minolta CM-2600d spectrophotometer (Konica Minolta, Tokyo, Japan), employing a D65/10° illuminant/observer configuration in SCI mode. Gloss was assessed using an Erichsen 503 instrument (Erichsen Cofomegra Instruments, Milan, Italy), following the ASTM D523/14 standard [63]. Additionally, the texture of the layers was studied with the MarSurf PS1 portable device for measuring surface roughness (Carl Mahr Holding, Gottingen, Germany).

To assess how the inclusion of bio-based fillers affects the long-term performance of the paint, two accelerated degradation tests were conducted to simulate exposure to harsh conditions.

To assess the coatings’ UV radiation durability, the specimens were positioned inside a UV-B chamber UV173 Box Co.Fo.Me.Gra (Co.Fo.Me.Gra, Milan, Italy), for a duration of 50 h, following the ASTM G154-16 standard [64]. The accelerated aging test utilized Co.Fo.Me.Gra UV-B fluorescent lamps (313 nm), with an irradiance of 0.71 W/m^2^ and a chamber temperature of 60 °C. FTIR infrared spectroscopy measurements, colorimetric analysis, and gloss assessments were employed to determine any potential deterioration in the layers. Potential chemical alterations within the polymer framework were investigated using a Varian 4100 FTIR Excalibur spectrometer (Varian, Santa Clara, CA, USA), through the acquisition of FTIR spectra. 

The impact of the fillers on the thermal resistance of the paint was assessed by subjecting the samples to extreme thermal fluctuations, which were simulated using the climatic chamber ACS DM340 (Angelantoni Test Technologies, Perugia, Italy). The exposure test was conducted in accordance with the UNI 9429 standard [65] and comprised 15 cycles, each consisting of:4 h at +50 °C and relative humidity < 30%;4 h at −20 °C;16 h at room temperature.

In order to avoid the undesired intake of moisture by the poplar wood base, the five untreated sides of the 40 × 40 × 2 mm^3^ specimens were enclosed using silicone. This sealing process was carried out prior to the experiments to ensure the integrity of the wood substrate. To track alterations in the coatings’ visual characteristics during the test, color analysis was carried out after every three exposure cycles in the climatic chamber. Similarly, the adhesion of the coatings was assessed by performing a cross-cut test, following the ASTM D3359-17 standard [66]. This test was conducted to determine whether the thermal cycles had caused any alterations in the coatings’ adhesion to the wood surface.

The impact of the combined action of the two green additives on the shielding features of the acrylic matrix was investigated by employing cold liquid resistance tests, following the GB/T 1733-93 standard [67]. In this study, filter paper was dipped into solutions containing 15% sodium chloride, 70% ethanol, detergent, and red ink, separately. Subsequently, the soaked filter paper was positioned on the surface of the coating and shielded with a glass lid. After 24 h, the glass cover and filter paper were taken away and any remaining liquid on the coating surface was eliminated. The resulting imprints and changes in color were assessed using color analysis, allowing for the evaluation of the coatings’ resilience to various chemical substances. Additionally, the water resistance of the paints was assessed by performing a liquid water absorption test following the guidelines outlined in the EN 927-5:2007 standard [68]. To prevent water absorption by the poplar wood substrate, the five untreated sides of the 40 × 40 × 2 mm^3^ panels were effectively sealed with silicone using the same method employed for the climatic chamber exposure. The samples were pre-conditioned at 65% relative humidity and 20 °C, after which they were placed to float in a container of water. The moisture absorption, quantified in grams per square meter (g/m^2^), was ascertained by tracking the growth in weight before the test and once a day until 96 h of testing was reached.

To assess the impact of the green fillers on the surface wettability of the coatings, contact angle measurements were conducted in accordance with the ASTM D7334-08 standard [69]. A Nikon 60 mm lens with an aperture of f/2.8 (Nikon Instruments Europe, Amstelveen, the Netherlands) was utilized for capturing macro pictures. The measurement of the contact angle was conducted using the NIS-Elements Microscope Imaging software (Windows Version 4.30.01). Droplets were generated via a syringe and dispersed from a distance of approximately 2 cm. After achieving proper focus on the droplet, a photograph was captured and the wetting angle was calculated using the imaging software. To ensure statistical validity, each sample underwent 10 measurements. This enabled a comprehensive analysis of the surface wettability properties.

Moreover, two tests were employed to evaluate the effect of the curcuma and wax on the mechanical properties of the acrylic coatings: the Buchholz hardness indentation test and the scrub test. The Buchholz hardness test was conducted using an Elcometer 3095 Buchholz Hardness Tester (Elcometer, Manchester, UK). The Buchholz Hardness Tester comprises a beveled disc indentation tool with a sharp edge inserted into a stainless steel block and it is used by applying a consistent testing force of 500 g for 30 s on the surface of the coating. The test followed the ISO 2815 standard [70] and entailed gauging the extent of the impression created by a standardized instrument. This measurement provided an indication of the coatings’ hardness. The scrub test was carried out employing an Elcometer 1720 Abrasion and Washability Tester (Elcometer, Manchester, UK), following the ISO 11998 standard [71]. The coatings’ resistance to abrasion was assessed by determining the reduction in mass of the samples every 250 cycles (operated at a frequency of 37 cycles per minute), totaling 1000 cycles. It is important to mention that, in contrast to the standard procedure, the test was carried out in a dry mode, omitting the use of a cleaning solution. This adjustment was made to prevent the test solution from permeating the polymeric matrix and the wood, which might have influenced the outcomes. Lastly, the structural changes induced by the abrasive process on the coatings were examined using the SEM. This enabled a more detailed investigation of the impact of the abrasion mechanism on the samples.

## 3. Results and Discussion

### 3.1. Additives and Coatings Structure

Figure 1 depicts the visual representation of the two variants of green materials incorporated into the waterborne acrylic bio-based paint. Curcuma (Figure 1a) is observed as irregularly shaped granules, typical of bio-based organic powders [42], with dimensions generally larger than 60 µm. Although the granule sizes vary significantly, they remain within the manufacturer’s specified range of 400 µm or less, as stated in the technical data sheet. Nevertheless, this characteristic does not pose any problems for employing curcuma powder as a novel color additive because it reveals favorable solubility in the employed bio-based paint. In comparison, the rice bran wax powders (Figure 1b) possess significantly smaller dimensions, measuring less than 15 µm. EDXS investigations revealed the organic composition of the curcuma granules, characterized by prominent peaks corresponding to carbon and oxygen. Furthermore, the powder was found to contain minor traces of magnesium, aluminum, phosphorus, sulfur, chlorine, potassium, calcium, and silicon. In contrast, the wax primarily consists of carbon (≈93 wt.%) and oxygen (≈7 wt.%). The EDXS spectra can be found in Appendix A.

Consequently, the two bio-based agents were incorporated into the waterborne acrylic paint using the procedure outlined in Table 1. This led to the creation of four sample series (TT, CC, CCT, and CCW) that were the focus of this study. The SEM images presented in the left section of Figure 2 exhibit the cross-sectional view of the samples, while the images in the right section offer an overhead view of the layers observed with the optical microscope. The objective of the former images, acquired after a brittle fracture process performed in liquid nitrogen, is to highlight the impact of both curcuma and rice bran wax on the paint application process, leading to possible alterations in the overall morphology and microstructure of the resulting coating.

The examination of samples TT and CC (Figure 2a and Figure 2b, respectively) reveals the perturbing influence of curcuma on the efficiency of the deposition process. Indeed, the two layers of sample TT, acquired using the unaltered acrylic paint, exhibit a leveling effect and form a coating with considerable thickness, averaging over 100 µm. In contrast, curcuma leads to a noticeable reduction in the coating thickness in sample CC. This occurrence can be attributed to a rise in the viscosity of the paint, leading to alterations to the spray process yield. Despite the substantial impact on the coating deposition, its structure remains compact and uniform. Simultaneously, the top-view images accentuate the impressive coloring potency of the bio-based pigment, which provokes significant changes in the visual aspect of the transparent paint, shifting towards yellow-orange hues. The addition of a third transparent layer in sample CCT (Figure 2c) leads to a substantial augmentation in the coating thickness, comparable to that of sample TT. Additionally, the third layer has the ability to smooth out the surface morphology of the coating without seemingly altering the aesthetic qualities imparted by curcuma in the first two layers. This is evident from the appearance of the sample in the top-view image. Finally, a similar behavior is observed in sample CCW (Figure 2d), whose third layer is added with wax. The cross-sectional figure accentuates the morphological distinction between the first two layers containing curcuma and the third layer containing the wax. The latter exhibits a greater thickness compared to the previous two layers, indicating a minimal impact on the process yield, similar to pure acrylic paint. However, it also demonstrates a more irregular internal structure due to the high concentration of bio-based fillers. Ultimately, the appearance of the coating closely resembles that of samples CC and CCT, indicating a minimal aesthetic impact of the wax, which does not significantly influence the transparency of the acrylic matrix.

To confirm these assumptions, surface roughness analyses were performed on the coatings, while their aesthetics were evaluated with gloss and colorimetric measurements. Table 2 provides a summary of the surface roughness values [Ra] for the samples, calculated by averaging 50 measurements taken from 10 samples in each, with 5 measurements collected from each individual sample. The assessments were taken both along the wood grain (//Ra) and across the wood grain (⊥Ra) directions. As the acrylic paint conforms to the texture of the wooden substrate, the transverse surface texture of sample TT is higher than the roughness in line with the fibers. Consistent with the SEM images, the CC sample, with its thin coating that adheres to the surface morphology of the wooden substrate, exhibits elevated Ra values in both the longitudinal and transverse orientations. The third layer of samples CCT and CCW effectively serves as a leveling agent, significantly reducing the elevated roughness values caused by the first two layers containing curcuma. However, it does not completely attain the smoothness achieved by the sole pure acrylic matrix (TT sample).

The variations in the roughness levels of the coatings have a notable impact on the appearance of the samples. In particular, an increase in Ra leads to a reduction in gloss, as depicted in Figure 3a. The graph illustrates the comparison of the gloss values of the coatings with those of the sample TT, which is used as a reference point. This comparison aims to emphasize the effect of the two green agents on the aesthetic features of the composite layers. In contrast to the gloss measurement of sample TT equal to 27.8, sample CC exhibits a gloss value of 3.9, indicating a significant reduction of 23.9 points. This decrease is directly associated with the elevated roughness values observed in the coating with added curcuma, as shown in Table 2. Sample CCT shows an increasing gloss value, equal to 13.8, thanks to the leveling power of the third layer of pure transparent acrylic matrix, which enhances the overall reflectance of the coating. However, the third layer of sample CCW sample does not possess the same reflective capabilities, with a gloss equal to 7.9. Despite the reduction in roughness of the composite film, the presence of the rice bran wax has a notable matting effect, typical of wax-based additives for coatings [72,73], reducing the capacity of the sample to efficiently refract light.

Similarly, the graph of Figure 3b emphasizes the color change ΔE of the composite coatings compared to sample TT, which can be attributed to the addition of the acrylic paint with the two bio-based fillers. ΔE was determined following the guidelines of the ASTM E308-18 standard [74]:ΔE = [(ΔL*)^2^ + (Δa*)^2^ + (Δb*)^2^]^1/2^, (1)
where the colorimetric coordinates L*, a*, and b* indicate the lightness (0 for black and 100 for white objects), the red-green axis (red for positive values while green is represented by negative values), and the yellow-blue coordinate (with positive values indicating yellow and negative values indicating blue), respectively. The addition of curcuma in the acrylic matrix of the paint causes a high color change ΔE in sample CC, equal to about 75 points. This phenomenon can be attributed to a notable decrease in L* and, more significantly, a substantial increase in b*. As a result, the sample undergoes a shift towards slightly darker tones, predominantly exhibiting shades of yellow. Taking into account the definition of the literature expressing a value of ΔE ≥ 1 as significant enough to be noticeable to the human eye [75,76], it can be concluded that curcuma exhibits a strong and impactful coloring ability. Furthermore, the optical microscope images presented in Figure 2 validate this remarkable coloring capability. The inclusion of an extra transparent layer in sample CCT does not result in any noticeable aesthetic alterations to the system, as the ΔE values remain comparable to those of sample CC. Similarly, the presence of rice bran in the top layer of sample CCW does not exert a notable influence on the color imparted by the underlying layers containing curcuma. As a result, the use of a wax-based filler allows for additional functionality of the coating without compromising the aesthetics of the system.

The conducted analyses reveal the impressive coloring ability of curcuma, resulting in a distinct yellow hue in the paint. The bio-based pigment significantly influences the coating’s morphology, leading to a considerable increase in roughness. On the other hand, the introduction of wax in an additional top layer impacts the surface structure of the specimen by diminishing both the texture and sheen of the coating. As a result, the utilization of both these green agents in a multilayer setup can significantly modify the layer’s appearance, influencing its color and light-reflecting characteristics.

### 3.2. Endurance of the Samples in Hostile Conditions

Curcuma and rice bran wax each serve unique purposes in altering the visual aspects and aesthetics of wood paint. However, the samples were subjected to accelerated deterioration trials, which encompassed exposure to UV-B radiation and significant temperature shocks, to examine how the two agents influence the polymer matrix’s durability over time.

#### 3.2.1. Exposure to UV-B Radiation

Figure 4 illustrates the results of the FTIR measurements carried out on the coatings and on the poplar support, both before and after undergoing a 50 h exposure to UV-B radiation. The wooden substrate exhibits identical peaks in both spectra, indicating consistency in its composition. The spectral region between 3400 and 3300 cm^−1^ stands for the vibrational stretching of the -OH group [77]. Additionally, the stretching region from 3000 to 2800 cm^−1^ represents the -CH group found in cellulose, hemicellulose, and lignin [78]. The peak observed at 1729 cm^−1^ refers to the stretching vibrations of the unconjugated C=O group and to the particular segments of the polymer chains found in the wood, like esters. [79]. The bands at 1592 cm^−1^ and 1460 cm^−1^ indicate the C=C benzene ring vibration of lignin and C-H deformation vibration, respectively [80]. Lastly, the two peaks observed at 1236 cm^−1^ and 1028 cm^−1^ correspond to the C-O stretching and the typical C-O-C stretching vibrations of cellulose [81], respectively. As a consequence of the exposure of the wooden panel to UV-B radiation, the magnitude of the peak at 1729 cm^−1^ increases, while the signal of the peak at 1236 cm cm^−1^ decreases. This observation suggests the deterioration of the wooden panel and changes to the chemical composition of cellulose [27].

On the other hand, the FTIR spectra of the four organic layers show a significant level of likeness, with the peaks arising from the wooden base entirely concealed by the signal from the polymeric matrix. Similarly, the existence of curcuma and rice bran wax in the layers cannot be distinguished in the FTIR analyses, as their signals are overshadowed by the spectrum of the acrylic paint. The specific peaks observed at 2923 cm^−1^ and 2850 cm^−1^ correspond to the stretching vibrations of CH_3_ and CH_2_ groups, respectively. The most prominent signal at 1725 cm^−1^ is associated with the stretching of carbonyl groups. The peaks observed at 1448 cm^−1^ and 1379 cm^−1^ are attributed to C-H bending vibrations. The broad peak spanning from 1270 cm^−1^ to 911 cm^−1^ represents the stretching vibration of C-O bonds (ester band). Lastly, the signals observed at 841 cm^−1^ and 751 cm^−1^ can be attributed to C-H and C-C-C vibrations, respectively. As the spectra of the four coatings exhibit minimal changes following UV-B radiation exposure, it can be inferred that the acrylic matrix displays remarkable resilience to photo-oxidative deterioration triggered by UV radiation [82,83].

Despite not being explicitly indicated in the FTIR analyses, the exposure to UV-B radiation significantly affects the visual appeal of the specimens, as depicted in Figure 5, illustrating the evolution of their aspect as a consequence of the accelerated degradation trial. The cellulose and lignin composition, primary constituents of poplar wood, undergoes substantial degradation. This deterioration is reflected in the marked yellowing of the wooden board.

Sample TT displays a comparable yellowing effect, as its coating is composed of a pure transparent acrylic matrix. Nevertheless, the presence of the paint seems to limit the intensity of the chemical and physical deterioration observed in the sample. Acrylic paints are recognized for their exceptional ability to withstand UV light [82,84] and the yellowing of sample TT is not caused by the deterioration of the paint layer but rather by the degradation of the poplar substrate. This assumption is supported by a previous study [27], which demonstrated that when transparent acrylic wood paints are used on surfaces that do not experience color shifts due to UV light, they do not display visual changes linked to possible polymer deterioration. Consequently, the paint does offer some relief from wood decay but it does not completely eliminate it, particularly under the demanding conditions of the accelerated degradation test. Furthermore, since the paint is see-through, it does not mask the color alteration brought about by UV-B radiation in the poplar substrate.

Likewise, the CC coating, which is pigmented with curcuma, shows signs of degradation in the poplar substrate. The paint, initially vibrant yellow in color, tends to become transparent after exposure to UV radiation. This occurrence can be ascribed to the process of photo-degradation susceptibility of the curcuma-based pigment [58,59,60]. The aesthetic degradation of the sample is slightly mitigated by the addition of the acrylic top layer in the CCT sample. However, this top layer fails to provide sufficient protection to the underlying layers against aggressive radiation. The presence of rice bran wax in sample CCW appears to slightly enhance the protective performance of the top layer. However, it should be noted that the bio-based filler is not particularly known for its resistance to UV radiation [41].

These observations are reinforced by the ΔE color change graph depicted in Figure 6a. The acrylic paint used in sample TT significantly reduces the yellowing effect on the wood. However, there is still a noticeable color change indicated by the ΔE value of about 9 points. Likewise, the coatings containing curcuma exhibit a pronounced value of ΔE, which can be attributed to both the deterioration of the wood beneath and the decay of the natural coloring substance. Compared to sample CC, the third layer manages to reduce the color change by about 5 and 11 points in samples CCT and CCW, respectively, underlining the significant role played by both the acrylic matrix and the rice bran wax. Nevertheless, the weakening of the curcuma-derived pigment leads to such a pronounced color change that the behavior of sample CCW cannot be compared to that of the coating TT, which consists of a pure unpigmented acrylic matrix.

The decay of the acrylic matrix in coating TT is correlated with a decrease of approximately 7 points in gloss, as depicted in Figure 6b. In contrast, the photo-degradation of curcuma in sample CC does not lead to notable shifts in gloss, as the sample already exhibits very low values prior to exposure to UV-B radiation. Otherwise, sample CCT reveals the decay of the acrylic top layer, as shown by a reduction in gloss. This reduction in gloss is comparable to that observed in sample TT, but of lesser intensity, considering that the initial gloss value of sample CCT is already lower. This phenomenon can be attributed to the higher roughness values observed in the tri-layer sample CCT. The surface roughness plays a significant part in determining the gloss of the coating, and the higher roughness in sample CCT contributes to a greater, unaltered impact on the overall gloss. Sample CCW, on the other hand, does not show appreciable change in gloss. In addition to greater roughness, the wax introduces a mattifying effect into sample CCW such as not undergoing a further decrease in gloss after being subjected to UV-B radiation.

In conclusion, the acrylic matrix exhibits positive chemical resilience to UV-B radiation, managing to moderately mitigate the photochemical breakdown in the wooden material. Nonetheless, owing to its see-through quality, the paint uncovers the wood’s underlying yellowing. In contrast, curcuma powder, being an organic compound, is notably prone to deterioration when subjected to UV-B radiation, leading to a decline in its visual appeal. Although curcuma powder shows promising possibilities as a pigment for paints, it is important to exercise caution and prevent direct sunlight to guarantee its long-lasting resilience and color performance. The inclusion of an additional top layer containing wax provides only a partial reduction in the aesthetic degradation of the coating. Consequently, it can be concluded that the curcuma-based pigment is not fully suitable for outdoor applications.

#### 3.2.2. Climatic Chamber Exposure

Table 3 outlines the assessment of the performance of the samples in the climatic chamber exposure test based on the formation of defects (cracks) and changes in their appearance (whitening phenomena), following the evaluation criteria established by the UNI 9429 standard [65].

Organic layers can be vulnerable to sudden temperature fluctuations because they tend to become fragile in cold conditions and are naturally prone to moisture penetration along their porous structure. These conditions promote the development of cracks [27,41], which can compromise the protective capabilities of the polymer layer. Hence, the existence of fissures is a pivotal aspect to take into account when assessing how samples react to thermal strains. In this context, all four series of samples exhibited positive performance, as their respective coatings displayed no visible signs of fractures. According to the standard, these four layers can be categorized as level 0, indicating the absence of defects, as confirmed through observations using an optical microscope at 4× magnification.

Resembling the classification of “cracks”, the samples demonstrate favorable behavior concerning the consistency of their appearance. The change in ΔE was tracked throughout their exposure in the climatic chamber and is illustrated in Figure 7. The presence of curcuma imparts a yellow hue to the paint, which is minimally affected by the thermal deterioration encountered during the experiment. The pigmented samples (CC, CCT, and CCW) exhibited a final color change of approximately 4 units. However, taking into account the organic and inherent characteristics of the pigment, this value of ΔE indicates a very minimal, almost negligible, alteration in the final color [38,41]. Regardless, the observed color change is not attributed to whitening effects since the L* values, the achromatic contributor, remain consistent throughout the test duration. However, there is a slight decrease in the a* and b* coordinates, resulting in the behavior depicted in the graph shown in Figure 7. This decrease accounts for the color change observed in the samples. The outputs of the CCT and CCW samples are practically comparable, highlighting the chromatic consistency of the rice bran wax. Consequently, the presence of this bio-based additive does not adversely affect the performance of the acrylic paint, even when exposed to various thermal cycles.

Following the completion of 15 thermal cycles, a cross-cut test was conducted on the samples to determine if the accelerated degradation test had caused any adhesion problems with the coatings. All four series of samples demonstrated outstanding adhesion even after undergoing multiple thermal changes, meeting the criteria of level 5B according to the ASTM D3359-17 standard [66]. This result can be confirmed by Figure 8, which shows no material removal on the surface of the coatings as a result of the cross-cut test. Once again, both the curcuma pigment and the rice bran wax-based filler had no detrimental impact on the preserving effectiveness of the acrylic matrix.

Overall, all the samples exhibit remarkable robustness when exposed to continuous temperature fluctuations, as shown by the slight whitening effects and absence of fractures in the coatings. This durability is further supported by the excellent adhesion maintained by the coatings, unaffected by the accelerated degradation test. The preservative properties of the polymer matrix remain largely unaffected by the inclusion of curcuma pigment and rice bran wax, which exhibit notable durability under thermal strains. Therefore, these agents have the potential for use in protective layers employed in environments characterized by significant temperature fluctuations.

However, it is recommended to refrain from exposing the composite coating directly to sunlight to maintain the color quality of the curcuma-based pigment. Solar radiation can lead to light-induced oxidation of the organic pigment, resulting in aesthetic decay phenomena of the paint.

### 3.3. Coatings Liquid Resistance

The liquid resistance test offers significant insights into the protective characteristics of wood coatings and the potential impact of pigments [28,38] and additives [37,85] incorporated into the paint. The assessment of the test results involves examining the level of color change occurring due to the interrelation of the coating and the test solutions, as indicated by the sources listed in Table 4 [86].

Figure 9 illustrates the results of the experiment, revealing the precise extent of hue modification caused by the four test solutions on the respective coatings. When considering the solutions of NaCl, ethanol, and detergent, sample TT exhibits exceptional performance, as evidenced by its ΔE values falling within the category 0 of the degree of fading. In contrast, curcuma experiences notable challenges when in contact with the test solutions, exhibiting significant discoloration effects, particularly with the detergent solution. This phenomenon is associated with the alkaline pH of the cleaning solution, equal to approximately 11. Certainly, it is widely recognized that curcumin’s water solubility is enhanced in alkaline environments, leading to a higher rate of chemical decomposition [87]. Consequently, the color of curcumin diminishes as a result of this reaction caused by alkaline degradation. The top layer of the CCT sample limits this undesired occurrence, reducing the extent of discoloration. Ultimately, the inclusion of rice bran wax (CCW sample) enhances the overall performance of the coating, leading to improved chromatic consistency. This behavior can be attributed to the inherent water-resistant characteristics of the wax, which efficiently reduce the absorption of watery solutions within the layer [88]. In summary, it can be concluded that the limited ΔE values indicate strong shielding characteristics of the multilayer composite films incorporating both curcuma and rice bran wax. On the contrary, when red ink is applied, it leads to a significant color alteration on the surface of the four samples, resulting in a color alteration rating of 5. This result was expected because of the distinctive traits of the strongly pigmented red ink; it is capable of quickly permeating the polymeric composition of the paints [37,41]. In this scenario, sample TT exhibits the most pronounced color change, primarily because it begins with lighter shades indicated by higher L* values. The addition of curcuma in the other three samples slightly “masks” the decrease in the L* coordinate while simultaneously increasing the a* coordinate. 

Indeed, samples CC and CCT, which initially have similar colors, display comparable behavior, indicating that the acrylic top layer has minimal impact in contrasting ink absorption. Conversely, sample CCW demonstrates a slight chromatic improvement, attributable to the barrier properties of the wax.

Overall, the test results indicate that the inclusion of curcuma has a negative impact on the isolation properties of the acrylic matrix, or at the very least, it promotes discoloration effects when the layer interacts with the mentioned test solutions. On the other hand, the presence of an additional top layer containing rice bran wax proves to compensate for the inadequate color resistance of curcuma, thereby enhancing the barrier performance of the system.

Nonetheless, it is important to highlight that the liquid resistance test exclusively yields qualitative outcomes linked to color alterations in the coatings. Hence, to evaluate the true numerical impact of the two bio-based additives on the barrier characteristics of the coatings, the samples were additionally examined using the liquid water absorption test. Figure 10 depicts the outcomes of the experiment, illustrating the progression of water absorption phenomena observed throughout the test. Moreover, the chart incorporates a line depicting the performance of the wooden base, used as a benchmark to underscore the insulating properties of the four groups of coatings. As expected, the wooden substrate demonstrates significantly greater water absorption over the duration of the test compared to the four coated samples. The three samples TT, CC, and CCT exhibit an almost identical behavior, indicating that curcuma does not have a substantial influence that promotes solution absorption within the acrylic matrix. Conversely, the inclusion of rice bran wax in the top layer of sample CCW results in a noticeable reduction in the quantity of solution absorbed by the sample. This finding further reinforces the positive outcomes observed in the previous liquid resistance test.

The unique hydrophobic properties of wax are widely recognized [89]. Nevertheless, to assess surface wetting properties and enhance comprehension of the wax’s hydrophobic function, contact angle measurements were conducted on the specimens. The outcomes of the experiment are outlined in Table 5 and Figure 11 provides a visual depiction of the measurement process, revealing the response of sample CCW.

The results suggest that curcuma does not significantly influence the hydrophobic/hydrophilic characteristics of the acrylic paint surface, as indicated by the comparable contact angle measurements of the TT, CC, and CCT samples. In contrast, the presence of rice bran wax in sample CCW proves to enhance the hydrophobic properties of the coating, as the average contact angle of the previous three samples is around 68°, while sample CCW exhibits an increased contact angle of 85°, representing an improvement of approximately 25%. Thus, this test once again reinforces the notion that the functionality of the wax-based filler can be exploited to effectively enhance the hydrophobicity of the wood paint. Numerous studies emphasize the role of surface roughness in altering the hydrophobic-hydrophilic characteristics of a surface [90,91]. An increase in surface roughness is commonly associated with an increase in surface hydrophobicity. Table 2 illustrates that the introduction of wax led to a slight rise in //Ra compared to the reference sample TT. However, it cannot be asserted that the enhanced hydrophobic behavior of the CCW sample is attributed to increased roughness. In fact, the Ra values are comparable to those of sample CCT and significantly lower than those of sample CC, which contains only curcuma. Upon comparing the roughness values in Table 2 with the contact angles in Table 5, it becomes evident that the CCW sample’s hydrophobicity is primarily a result of adding rice bran wax, rather than morphological aspects of its surface.

In summary, it can be concluded that curcuma does not promote specific percolation and solution absorption phenomena within the acrylic matrix of the coating. However, its vulnerability to color degradation poses a significant challenge when the coating interacts with different solutions. To address these concerns, the inclusion of rice bran wax has demonstrated significant success in enhancing the barrier capabilities of the coating, reducing color fading, and enhancing the hydrophobic properties of the system. As a result, these findings offer insights into potential novel uses of these two naturally derived agents in multilayer films, where their synergistic effects can be harnessed to impart distinctive aesthetic aspects and excellent resistance to liquids.

### 3.4. Coatings Mechanical Features

Figure 12 displays the results of the Buchholz hardness test. Each column, representative of the mean size of the indentations, is associated with the corresponding Buchholz hardness value. 

The hardness characteristics of samples TT, CC, and CCT are comparable to each other. It should be noted that curcuma was added primarily for coloring purposes and not specifically to enhance the mechanical properties of the acrylic paint. However, the inclusion of rice bran wax offers a modest strengthening effect, as highlighted in the figure by a reduction in the size of the indentations with the incorporation of green additives (sample CCW). The wax seems to lead to a significant decrease in the length of the indentations, by approximately 8%. Nevertheless, the final hardness values for all four sample series fall within the <50 range defined in the standard [70], as the lengths of the indentations exceed 2000 µm in length (defined as the maximum value admissible by the test). Hence, the wax does not play a significant role in causing a substantial rise in the paint’s hardness. Indeed, rice bran wax has relatively high melting points ranging from 78 to 81 °C, yet it remains pliable and ductile at room temperature [92]. Thus, this good ductility could be the cause of a return of the deformation of the coating caused by the indenter, with consequent distortion of the notch measurement.

Nonetheless, these pliable characteristics can be harnessed to create surfaces that possess improved resistance to abrasion. For instance, previous studies have investigated the application of wax-based additives to create slip-resistant surfaces [41,93]. Figure 13 showcases the development of mass reduction in the four coatings during the scrubbing test. As seen in the earlier hardness examination, the performance of sample CC is akin to that of the coatings TT and CCT, since curcuma does not contribute to the reinforcement of the acrylic paint. However, the bio-based pigment does not negatively impact the mechanical durability of the polymeric matrix either. In this regard, curcuma serves as a superb pigment, delivering exceptional coloring qualities without jeopardizing the safeguarding properties of the polymer matrix. On the other hand, the introduction of rice bran wax in the top layer of sample CCW yields the desired effect by significantly decreasing the weight loss throughout the abrasive process. This phenomenon becomes evident in the initial monitoring after 250 cycles and intensifies as the test progresses. After 1000 scrub cycles, the CCW sample demonstrates a notable reduction in mass loss, approximately ranging from 33% to 36%, compared to the other three samples.

Nevertheless, to further assess the strengthening effect of the rice bran wax, the specimens were subjected to SEM inspections upon the completion of the abrasion test. For instance, Figure 14 offers a side-by-side view of the surfaces of samples TT and CCW, analyzed using secondary electron mode. The images emphasize the existence of abrasion lines and the harm inflicted on the coatings’ surfaces by the scrubbing pad. This imperfection, coupled with a corresponding material loss, seems to be more frequent on the surface of sample TT, which shows a particularly jagged surface. Consequently, the images suggest that sample CCW exhibits superior resistance to abrasion compared to sample TT, as previously revealed in Figure 13. Although the surface of sample TT exhibits a significantly uneven texture, which is typical of the common imperfections caused on organic coatings by the abrasive action of the scrubbing pad [72,94,95,96], the surface of sample CCW presents a distinctly contrasting appearance. The specific irregularities resulting from abrasive processes observed in sample TT are substituted by a surface featuring notably flat regions, subjected to particular plastic deformation phenomena. The movement of the scrubbing pad does not lead to matter eradication; instead, it induces the plastic deformation of the rice bran wax, known for its malleability even at room temperature [92]. This phenomenon is typical of wax-based additives utilized in the creation of slick surfaces [93]. The pad tends to “disperse” the wax, which in turn lessens the friction between the coating surface and the pad itself [41]. As a result, the strengthening impact of the wax arises from its ability to undergo plastic deformation and alleviate surface friction effects. Consequently, the coating encounters a reduced degree of material loss, as it is not entirely removed by the pad but instead distributed across the composite layer’s surface.

These events impact the surface texture of the coatings, as demonstrated in Table 6, which summarizes the changes in perpendicular roughness (⊥Ra) values as a result of the scrubbing test cycles. The wear and matter subtraction process led to a noticeable rise in roughness for samples TT, CC, and CCT, corroborating the distinct irregular surface morphology depicted in Figure 14a and corroborating the analogous behavior observed in Figure 13. Otherwise, the values of ⊥Ra of sample CCW remain practically unchanged, since the flattening of the wax does not lead to an increase in roughness.

In summary, the curcuma-based additive does not undermine the preservative effectiveness of the polymeric matrix. Conversely, the rice bran wax offers a noteworthy contribution that does not considerably enhance the coating’s hardness but effectively prevents mechanical deterioration phenomena caused by abrasion stresses. Therefore, curcuma and rice bran wax act synergistically, with the former providing vibrant and distinct yellow coloring in the first two layers of sample CCW, while the latter reinforces the abrasion resistance of the top layer of the coating.

## 4. Conclusions

This study emphasizes the diverse qualities exhibited by curcuma-based pigment and rice bran wax; both of which have been selectively utilized as bio-based additives in the development of multifunctional multilayer wood coatings. The investigations conducted in this study unveiled the remarkable coloring properties of curcuma, which imparted a unique yellow shade to the paint. Furthermore, the incorporation of rice bran wax in an additional top layer had an impact on the surface characteristics of the samples, leading to a reduction in both the roughness and gloss of the coating. 

Upon exposure to UV-B radiation, the coating exhibited a noticeable and consistent color alteration due to significant photo-degradation of the curcuma powder. However, the preservative properties of the polymeric matrix remained unaffected by the presence of both curcuma and rice bran wax additives, demonstrating their remarkable stability under thermal stresses. 

Furthermore, the tests conducted to assess liquid resistance demonstrated that neither curcuma nor rice bran wax had a detrimental impact on the shielding capabilities of the paint. The additives did not introduce significant discontinuities in the acrylic matrix, thereby preventing excessive liquid absorption. Additionally, the inclusion of wax in the coatings imparted effective water repellence, indicating an enhancement in the barrier properties of the layers. Crucially, the durability of the paint remained uncompromised by the curcuma pigment and the wax agent had a strengthening effect, also mitigating the potential mechanical deterioration from abrasive stresses.

Therefore, the combination of curcuma and rice bran wax offers intriguing possibilities as bio-based additives for wood coatings. Their application results in vibrant coloring and specific aesthetic qualities in the paint. Additionally, it boosts the hydrophobic characteristics of the coated surface and enhances the composite layer’s resilience to abrasion.

## Figures and Tables

**Figure 1 materials-16-07086-f001:**
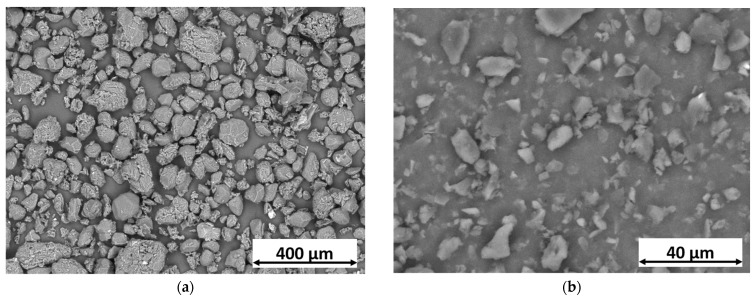
SEM micrographs of (**a**) curcuma granules and (**b**) rice bran wax, acquired with backscattered electrons (BSE).

**Figure 2 materials-16-07086-f002:**
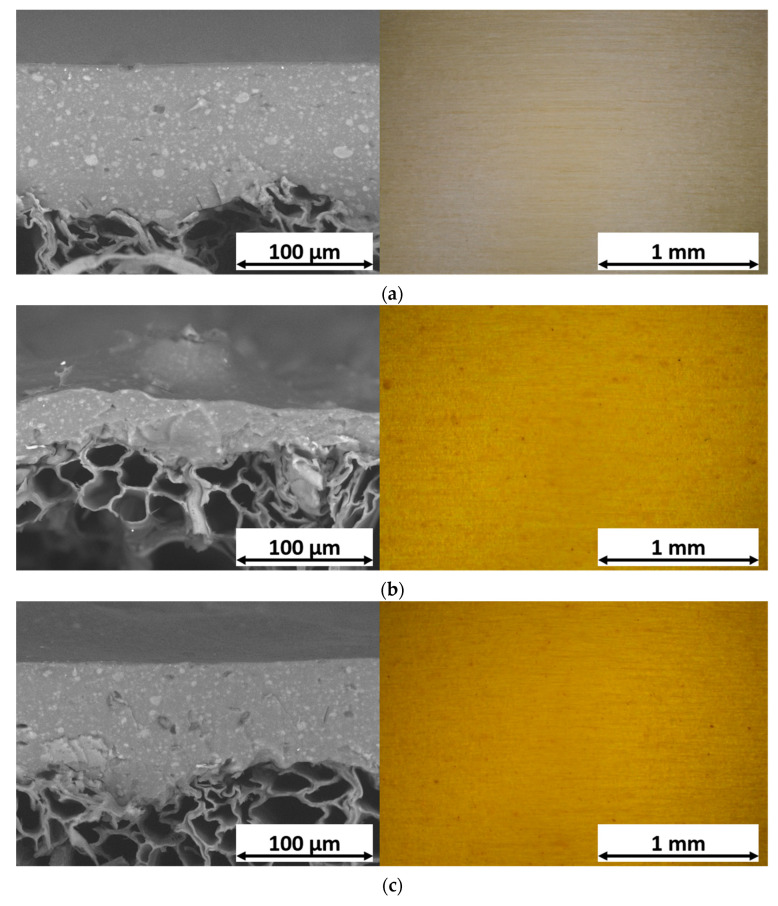
SEM micrographs of the cross-section (on the left) and top-view optical microscope micrographs (on the right) of (**a**) sample TT, (**b**) sample CC, (**c**) sample CCT, and (**d**) sample CCW.

**Figure 3 materials-16-07086-f003:**
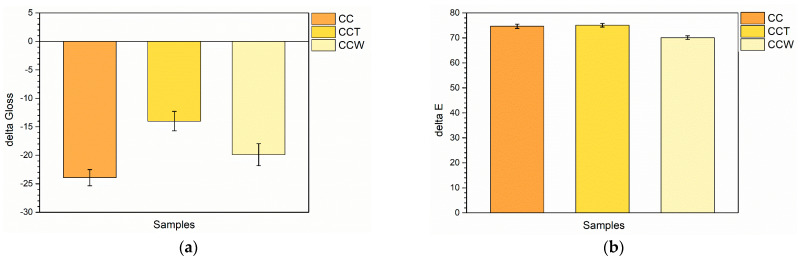
Variation of (**a**) gloss and (**b**) color compared to sample TT.

**Figure 4 materials-16-07086-f004:**
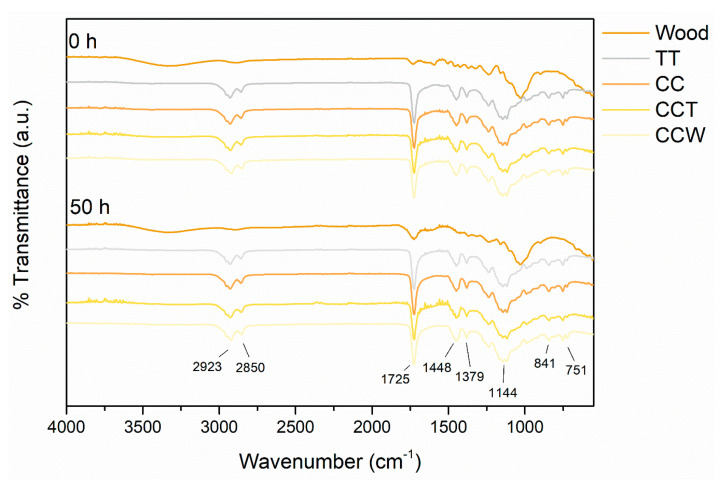
Evolution of the FTIR spectra of the samples before and after the exposure to UV-B radiation.

**Figure 5 materials-16-07086-f005:**
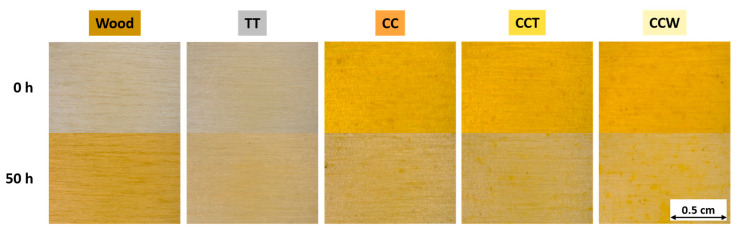
Modification of the appearance of the wooden panel and the coated samples as a consequence of the exposure to UV-B radiation.

**Figure 6 materials-16-07086-f006:**
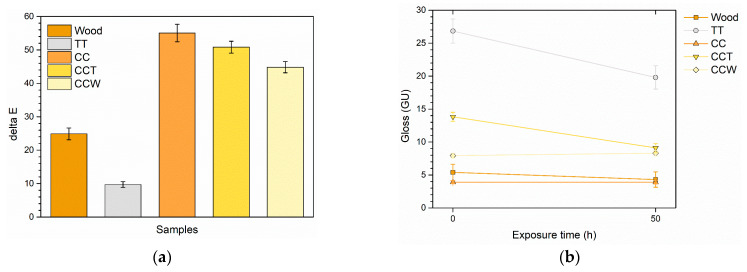
Evolution of (**a**) color and (**b**) gloss during UV-B exposure.

**Figure 7 materials-16-07086-f007:**
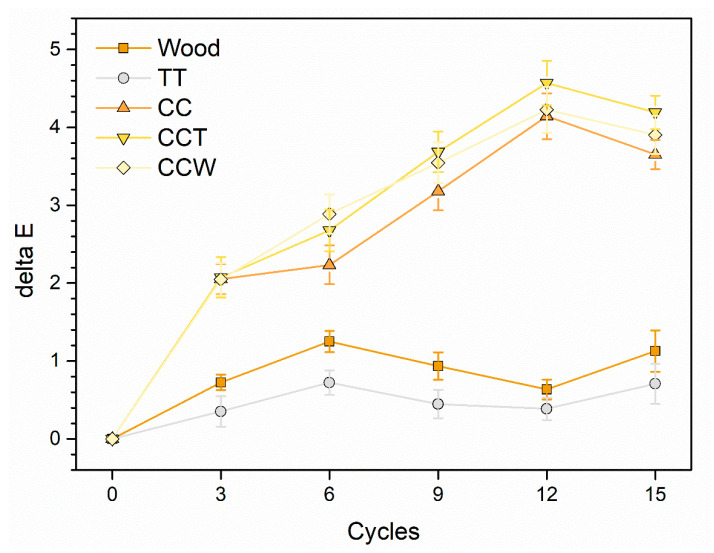
Color evolution induced by climatic chamber exposure.

**Figure 8 materials-16-07086-f008:**
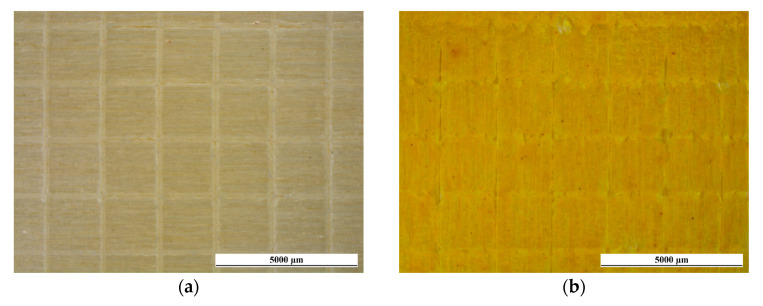
Cross-cut test results after exposure in the climatic chamber of (**a**) sample TT, (**b**) sample CC, (**c**) sample CCT, and (**d**) sample CCW, observed with an optical microscope.

**Figure 9 materials-16-07086-f009:**
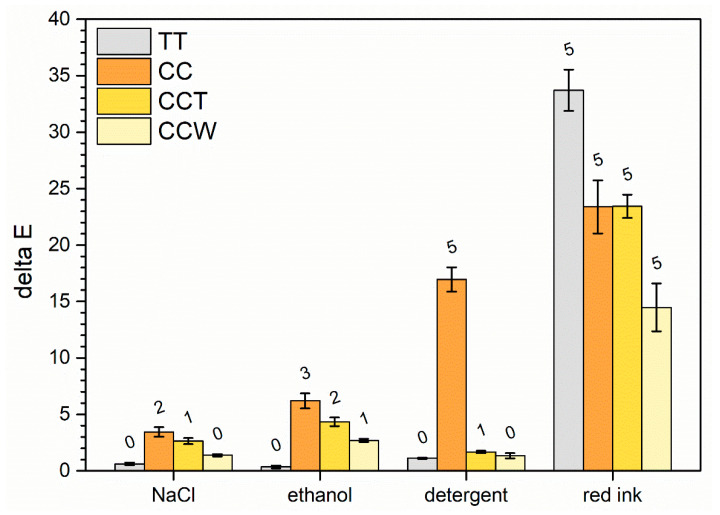
Alterations in the color of the coatings following exposure to liquids. The numerical values displayed above the columns represent the levels of discoloration recorded in Table 4.

**Figure 10 materials-16-07086-f010:**
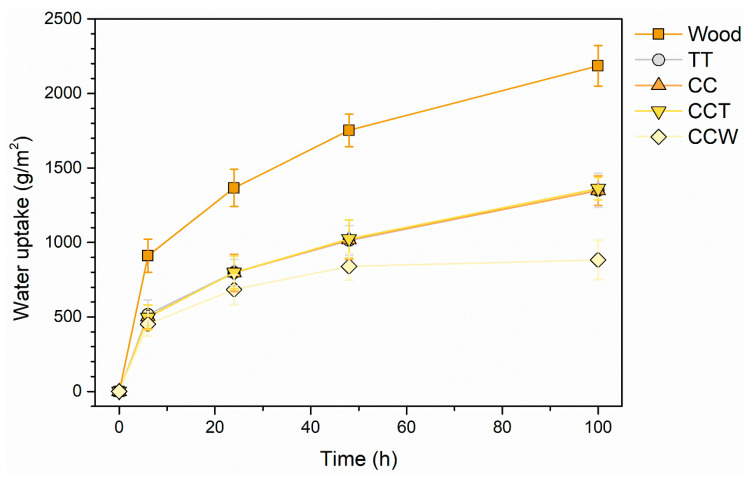
Water absorption evolution throughout the experiment.

**Figure 11 materials-16-07086-f011:**
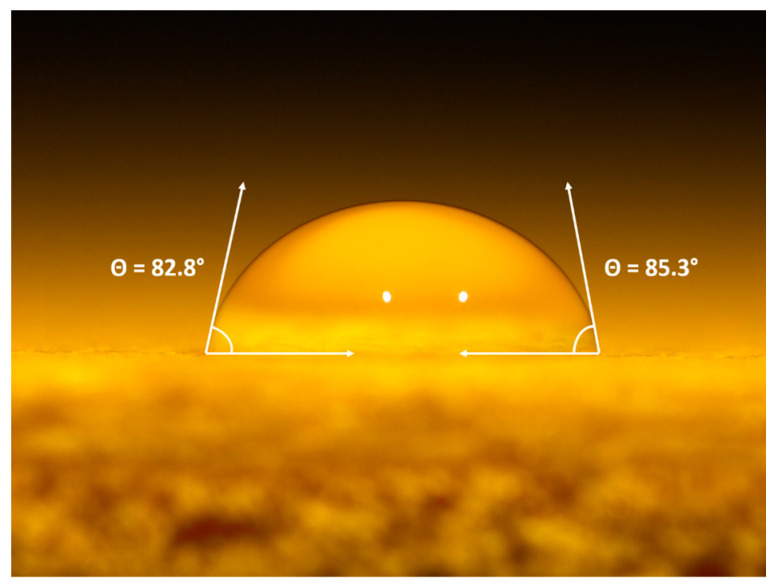
Optical micrograph of the contact angle measurements of sample CCW.

**Figure 12 materials-16-07086-f012:**
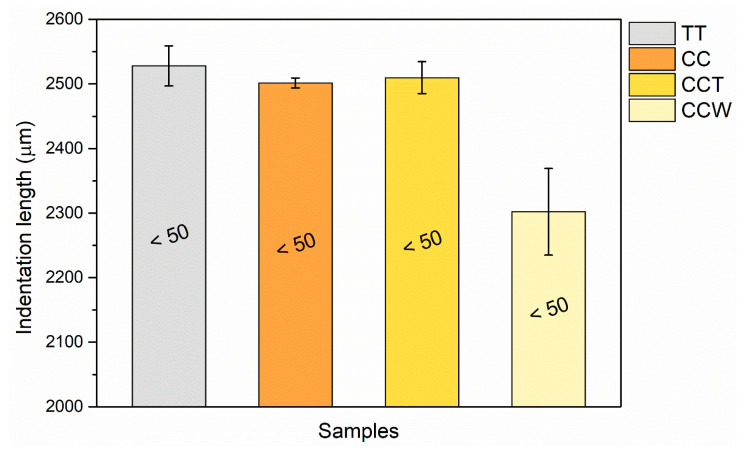
Average indentation length of the notches of the Buchholz test, alongside the corresponding Buchholz hardness values.

**Figure 13 materials-16-07086-f013:**
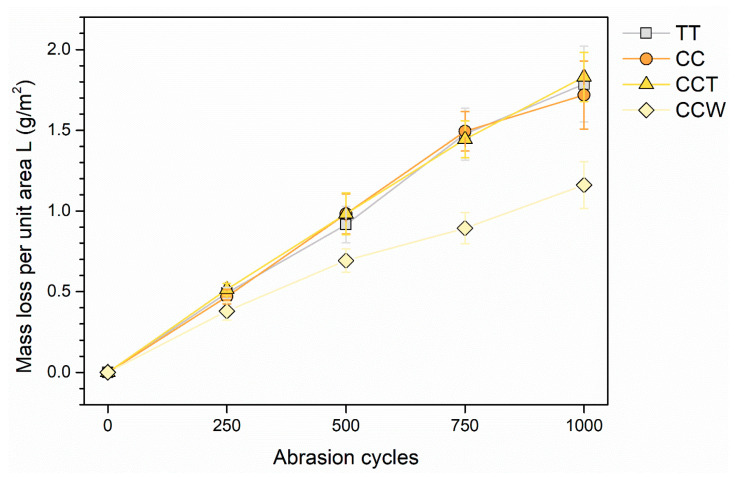
Loss of coating mass per unit area, depending on the number of scrub abrasion cycles.

**Figure 14 materials-16-07086-f014:**
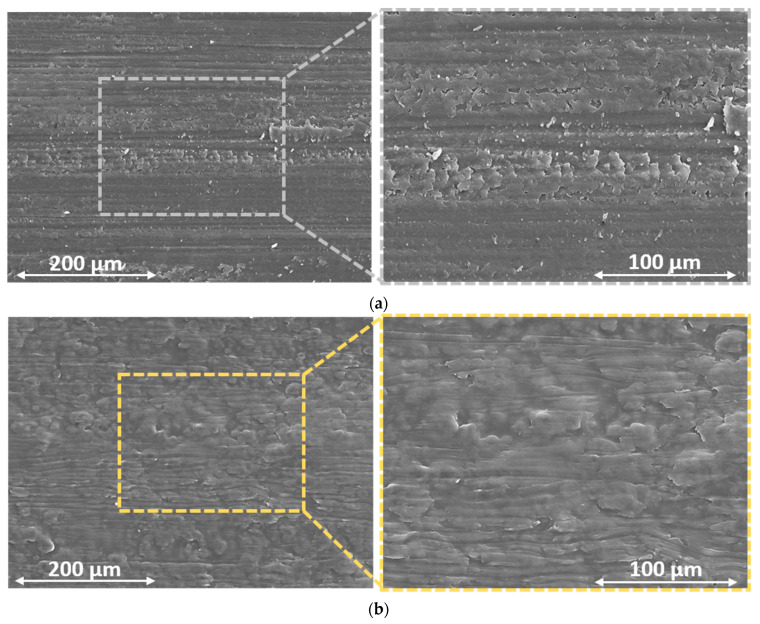
Comparison of the top-view SEM micrographs of the surface morphology of (**a**) sample TT and (**b**) sample CCW after the 1000 scrub test cycles.

**Table 1 materials-16-07086-t001:** Samples nomenclature with corresponding bio-based filler additivation.

Samples Nomenclature	First Layer	Second Layer	Third Layer
TT	/	/	
CC	Curcuma (5 wt.%)	Curcuma (5 wt.%)	
CCT	Curcuma (5 wt.%)	Curcuma (5 wt.%)	/
CCW	Curcuma (5 wt.%)	Curcuma (5 wt.%)	Rice bran wax (10 wt.%)

**Table 2 materials-16-07086-t002:** Coatings surface roughness.

Sample	Longitudinal Roughness //Ra (µm)	Transverse Roughness ⊥Ra (µm)
TT	0.51 ± 0.03	2.24 ± 0.14
CC	7.58 ± 2.08	9.18 ± 0.69
CCT	2.34 ± 0.43	2.89 ± 0.33
CCW	2.24 ± 0.61	1.96 ± 0.31

**Table 3 materials-16-07086-t003:** Classification of crack development and whitening phenomena resulting from the climatic chamber exposure.

Category	Defects	Whitening
0	No alterations	No whitening
1	Defects only discernible using a 4× optical system	Light whitening
2	Evident cracks	Intense whitening

**Table 4 materials-16-07086-t004:** Values representing the magnitude of color alteration associated with the degree of fading.

Level	Degree of Fading	Color Change ΔE
0	unchanged color	≤1.5
1	extremely slight coloration	1.6–3.0
2	small color shift	3.1–6.0
3	clearly visible fading	6.1–9.0
4	considerable alteration in color	9.1–12.0
5	complete loss of color	>12.0

**Table 5 materials-16-07086-t005:** Contact angle measurements (θ).

Sample	Θ (°)
TT	69.9 ± 2.8
CC	68.7 ± 5.5
CCT	66.6 ± 6.0
CCW	85.1 ± 5.2

**Table 6 materials-16-07086-t006:** Coatings surface roughness ⊥Ra before and after the scrub test.

Sample	⊥Ra (µm) before the Scrub Test	⊥Ra (µm) after the Scrub Test
TT	2.24 ± 0.14	3.56 ± 0.80
CC	9.18 ± 0.69	9.72 ± 0.84
CCT	2.89 ± 0.33	3.67 ± 0.89
CCW	1.96 ± 0.31	2.04 ± 0.21

## Data Availability

The data presented in this study are available on request from the corresponding author. The data are not publicly available due to the absence of an institutional repository.

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
