# Peer review of "Eco-Friendly Multilayer Coating Harnessing the Functional Features of Curcuma-Based Pigment and Rice Bran Wax as a Hydrophobic Filler"

_materials, 2023, doi:10.3390/ma16227086_

Round 1

Reviewer 1 Report

Comments and Suggestions for Authors

In this paper, Eco-friendly multilayer coating harnessing the functional features of curcuma-based pigment and rice bran wax as a hydrophobic filler (Manuscript ID: materials-2692316), in which the effect of the two additives, i.e., the curcuma-based pigment and rice bran wax on the morphology of the coatings and the influence on the durability of the paint were examined. Here are some comments that may help improve this manuscript as follows,

1.        The achievement, finding, or novelty of this work should be presented and highlighted in the abstract with representative property enhancement. The abstract should be summarized and simplified to some major findings or important outcome with specific data supporting.

2.        It was stated in line 86-90 that “Rice bran wax is composed of renewable hydrocarbon compounds and it is obtained through a dewaxing process during the refinement of rice bran oil [52,53], which includes degumming, deacidification, bleaching, deodorization, and winterization [53,54]. Due to its inherent hydrophobic properties, rice bran wax has the potential to serve as a plentiful and cost-effective bio-based filler, enhancing the protective capabilities of organic coatings.” Supportive references should be cited to enhance the statement that the hydrophobicity of rice bran wax is derived from the carbon-based component, which could be proved from the hydrophobicity of carbon materials (https://doi.org/10.1016/j.jallcom.2023.172582).

3.        It was stated in line 602-606 that “The unique hydrophobic properties of wax are widely recognized [86]. Nevertheless, in order to evaluate the surface wettability and gain a better understanding of the hydrophobic role of the wax, contact angle measurements were performed on the samples. The results of the wettability test are presented in Table 5, and Figure 11 displays an image captured during the measurements, showcasing the behavior of sample CCW.”, as indicated in table 5 and figure 11, the contact angle increases from 66 to 85°, which could be attributed to the effect of rice bran wax. While it should be noted that the enhancement in hydrophobicity could be due to the coupled effects of surface roughness (including the surface topology) and surface physicochemical property. While whether the introduction of rice bran wax enhances the surface roughness or surface chemical composition should be implied by further analysis with supportive references (https://doi.org/10.1016/j.jpowsour.2022.231121; https://doi.org/10.1016/j.jpowsour.2023.233077).

4.        In section 3.4, the hardness test should be well introduced and the corresponding indentation profile should be offered. The relationship between the indentation length and the hardness should be well introduced.

5.        The conclusion should be clear, concise, and conclusive, rather than a wordy and massive repeat of the result content.

Comments on the Quality of English Language

minor revision

Author Response

  1. The achievement, finding, or novelty of this work should be presented and highlighted in the abstract with representative property enhancement. The abstract should be summarized and simplified to some major findings or important outcome with specific data supporting.

Authors: The abstract has been modified, aiming to highlight the novelty of the work, which utilizes two bio-based fillers in wood coatings, leveraging their diverse features and eco-friendly nature. Additionally, the main findings of the study have been introduced in the abstract without providing specific data. The authors, in fact, prefer to refrain from including data in the abstract to avoid further lengthening it.

  1. It was stated in line 86-90 that “Rice bran wax is composed of renewable hydrocarbon compounds and it is obtained through a dewaxing process during the refinement of rice bran oil [52,53], which includes degumming, deacidification, bleaching, deodorization, and winterization [53,54]. Due to its inherent hydrophobic properties, rice bran wax has the potential to serve as a plentiful and cost-effective bio-based filler, enhancing the protective capabilities of organic coatings.” Supportive references should be cited to enhance the statement that the hydrophobicity of rice bran wax is derived from the carbon-based component, which could be proved from the hydrophobicity of carbon materials (https://doi.org/10.1016/j.jallcom.2023.172582).

Authors: The authors added the following sentence at line 106: “The low surface free energy of wax [55,56] can indeed be harnessed to enhance the hydrophobic properties of wood, as suggested by recent studies [57].” 3 new references have been introduced to support the analyses. However, the authors decided not to add the reference suggested by the reviewer, because it was not very relevant to the properties of the wax.

  1. It was stated in line 602-606 that “The unique hydrophobic properties of wax are widely recognized [86]. Nevertheless, in order to evaluate the surface wettability and gain a better understanding of the hydrophobic role of the wax, contact angle measurements were performed on the samples. The results of the wettability test are presented in Table 5, and Figure 11 displays an image captured during the measurements, showcasing the behavior of sample CCW.”, as indicated in table 5 and figure 11, the contact angle increases from 66 to 85°, which could be attributed to the effect of rice bran wax. While it should be noted that the enhancement in hydrophobicity could be due to the coupled effects of surface roughness (including the surface topology) and surface physicochemical property. While whether the introduction of rice bran wax enhances the surface roughness or surface chemical composition should be implied by further analysis with supportive references (https://doi.org/10.1016/j.jpowsour.2022.231121; https://doi.org/10.1016/j.jpowsour.2023.233077).

Authors: The authors thank the reviewer for the interesting food for thought. The authors added the following text to line 673, inserting the two recommended references: “Numerous studies emphasize the role of surface roughness in altering the hydrophobic-hydrophilic characteristics of a surface [90,91]. An increase in surface roughness is commonly associated with an increase in surface hydrophobicity. Table 2 illustrates that the introduction of wax led to a slight rise in //Ra compared to the reference sample TT. However, it cannot be asserted that the enhanced hydrophobic behavior of the CCW sample is attributed to increased roughness. In fact, the Ra values are comparable to those of sample CCT and significantly lower than those of sample CC, which contains only curcuma. Upon comparing the roughness values in Table 2 with the contact angles in Table 5, it becomes evident that the CCW sample's hydrophobicity is primarily a result of adding rice bran wax, rather than morphological aspects of its surface.” The comparison between Table 2 and Table 5 highlights a poor contribution of the surface morphology and the important impact of the wax in increasing the hydrophobicity of the coating.

  1. In section 3.4, the hardness test should be well introduced and the corresponding indentation profile should be offered. The relationship between the indentation length and the hardness should be well introduced.

Authors: The authors added the sentence at line 275: “The Buchholz Hardness Tester comprises a beveled disc indentation tool with a sharp edge inserted into a stainless steel block, ad it is used applying a consistent testing force of 500 grams for 30 s on the surface of the coating.” Moreover, the authors added the sentence at line 708: “Nevertheless, the final hardness values for all four sample series fall within the <50 range defined in the standard [70], as the lengths of the indentations exceed 2000 µm in length (defined as the maximum value admissible by the test).” The two sentences explain how the indenter is structured and define a relationship between the length of the indentation and the hardness value, according to the standard.

  1. The conclusion should be clear, concise, and conclusive, rather than a wordy and massive repeat of the result content.

Authors: The authors followed the suggestion of the reviewer, reducing the length of the Conclusions, avoiding repetitions, in order to highlight the positive results of the study.

Reviewer 2 Report

Comments and Suggestions for Authors

The manuscript reports interesting results. The manuscript is well written and structured. The characterization is up to date. I suggest minor comments: 

- Figure 3: please remove the graduation of the Y2-axis (right-hand side). Same for Figures 6, 7, 9, 10, 12, 13. 

- Figure 4: X2 graduation should be removed. Same for Figures 10, 13. 

- Figure 11: Why do the contact angles differ from one side? This should be commented on at the molecular level. 

- Figure 5: The scale should go inside the figure like Figures 1, 2, 8, and 12.

Comments on the Quality of English Language

The manuscript is good enough to be considered. 

Author Response

- Figure 3: please remove the graduation of the Y2-axis (right-hand side). Same for Figures 6, 7, 9, 10, 12, 13.

Authors: The images have been edited, as suggested by the reviewer.

- Figure 4: X2 graduation should be removed. Same for Figures 10, 13.

Authors: The images have been edited, as suggested by the reviewer.

- Figure 11: Why do the contact angles differ from one side? This should be commented on at the molecular level.

Authors: the contact angles differ mainly due to the presence of rice bran wax. The authors added the sentence at line 106, underlying the hydrophobic role of wax: “The low surface free energy of wax [55,56] can indeed be harnessed to enhance the hydrophobic properties of wood, as suggested by recent studies [57].” Moreover, the authors added the sentences at line 677: “Numerous studies emphasize the role of surface roughness in altering the hydrophobic-hydrophilic characteristics of a surface [90,91]. An increase in surface roughness is commonly associated with an increase in surface hydrophobicity. Table 2 illustrates that the introduction of wax led to a slight rise in //Ra compared to the reference sample TT. However, it cannot be asserted that the enhanced hydrophobic behavior of the CCW sample is attributed to increased roughness. In fact, the Ra values are comparable to those of sample CCT and significantly lower than those of sample CC, which contains only curcuma. Upon comparing the roughness values in Table 2 with the contact angles in Table 5, it becomes evident that the CCW sample's hydrophobicity is primarily a result of adding rice bran wax, rather than morphological aspects of its surface.” Therefore, the different hydrophobic behavior of the samples is due to the rice bran wax, which has low surface energy, rather than to the surface morphology of the coatings.

- Figure 5: The scale should go inside the figure like Figures 1, 2, 8, and 12.

Authors: The images have been edited, as suggested by the reviewer.

Round 2

Reviewer 1 Report

Comments and Suggestions for Authors

Capable of being further processed in current version.

Comments on the Quality of English Language

fine